# Progress towards Measles and Rubella Elimination in the South-East Asia Region—2013–2023

**DOI:** 10.3390/vaccines12101094

**Published:** 2024-09-25

**Authors:** Sudhir Khanal, Vinod Bura, Lucky Sangal, Raman Sethi, Deepak Dhongde, Sunil Kumar Bahl

**Affiliations:** 1WHO Regional Office for South-East Asia, New Delhi 110002, India; burav@who.int (V.B.); sanagllu@who.int (L.S.); sethir@who.int (R.S.); dhongded@who.int (D.D.); 2Public Health Expert (Formerly with WHO Regional Office for South-East Asia), New Delhi 110002, India; bahlsk@gmail.com

**Keywords:** measles, rubella, vaccination, elimination, vaccine preventable diseases, SEA Region

## Abstract

The South-East Asia (SEA) Region of the World Health Organization (WHO), through a Regional Committee resolution in 2013, adopted the goal of “measles elimination and rubella control by 2020”. The goal was revised in 2019 to “measles and rubella elimination by 2023”. Countries of the Region have made significant efforts to achieve the goal. Progress has been made in the Region, with five of the 11 countries of the Region having been verified for having eliminated measles and rubella. Surveillance and immunization program performance for measles and rubella has shown an improvement since 2013. This progress has been possible due to a high level of political and programmatic commitment in the countries of the Region, as well as due to the alliances and infrastructures established for disease elimination initiatives in the past, notably for polio, being utilized effectively to implement strategies for measles and rubella elimination. The unforeseen COVID-19 pandemic had a detrimental effect on the immunization and surveillance efforts, leading to a delay in the achievement of measles and rubella elimination in the Region. Challenges to achieve the goal remain; however, efforts are ongoing in countries to not only protect the gains made so far but also to make further progress towards the goal of measles and rubella elimination.

## 1. Introduction

The WHO South-East Asia Region (WHO South-East Asia Region has eleven countries—Bangladesh, Bhutan, DPR Korea, India, Indonesia, the Maldives, Myanmar, Nepal, Sri Lanka, Thailand, and Timor-Leste) adopted the goal of measles elimination (absence of endemic measles cases for a period of ≥12 months) and rubella and congenital rubella syndrome control (ninety-five percent reduction in disease incidence from the 2013 level) by 2020 through a Regional Committee resolution in 2013 [1,2]. An independent review of the progress towards rubella control in the Region was conducted in 2019, which recommended countries of the Region take advantage of the opportunity provided by measles elimination activities to revise the goal to include rubella elimination [2]. Following the recommendation, in the subsequent regional committee meeting of the South-East Asia Region held in September 2019, countries resolved to eliminate both measles and rubella by 2023 [3]. The strategies adopted by countries of the Region to eliminate measles and rubella included: (a) achieving high population immunity with more than 95% coverage of two doses of measles and rubella-containing vaccine through routine immunization, periodic intensification or supplementary immunization campaigns; (b) ensuring a sensitive laboratory supported case-based fever and rash surveillance system; (c) maintaining an accredited laboratory network to support fever and rash surveillance; (d) identifying, investigating and responding to measles and rubella outbreaks in a timely manner; and (e) linking these initiatives with other public health interventions to ensure optimal implementation of the four strategies listed above [4].

This article presents a review of progress towards measles and rubella elimination and the changes in epidemiology of the diseases over time in the WHO South-East Asia Region, the factors determining the progress, and the remaining challenges.

## 2. Methods and Materials

The progress towards measles and rubella elimination was measured between 2013 and 2023.

Cessation of transmission, disease incidence, reported cases, mortality information (only for measles), surveillance performance across various indicators (for the last 3 years), and epidemiological analysis were used to measure progress.

Published data from the WHO-UNICEF estimates of national immunization coverage were used to review progress towards vaccination coverage, while the estimated number of measles fatalities and cases prevented over this time was taken from WHO published articles on global progress towards measles and rubella elimination [5].

The change in epidemiology of measles and rubella during the last 3 years was reviewed for changes in age distribution and vaccination status using the data from case-based surveillance for suspected measles and rubella submitted by countries of the Region to WHO through the joint reporting form as well as weekly reports [6,7].

Data on the changing genotype of measles and rubella since last 10 years are also presented using the measles and rubella nucleotide surveillance database—MeaNS and RubeNS, respectively [8,9].

The reports from the Regional Verification Commission for Measles and Rubella Elimination (SEA-RVC) [10,11,12,13] and the reports from the Regional Immunization Technical Advisory Group (SEAR-ITAG) [14,15,16,17] were reviewed for information needed to identify the timelines and the challenges related to program implementation.

## 3. Results

### 3.1. Cessation of Transmission

In 2013, all the countries in the South-East Asia Region of WHO were endemic to measles and rubella. As of 2023, five of the eleven countries in the Region—Bhutan, DPR Korea, the Maldives, Sri Lanka, and Timor-Leste—have been verified by the SEA-RVC to have eliminated measles and rubella, based on the verification guidelines developed by WHO [18,19].

### 3.2. Disease Incidence, Reported Cases and Mortality

The reported incidence for measles cases in the SEA Region increased from 16.2 per million population in 2013 to 40.9 per million in 2023. Only four countries experienced a decline in incidence, with significant increases in incidence reported in India and Indonesia. The reported incidence of rubella cases in the Region decreased from 5.5 per million population in 2013 to 1.4 per million population in 2023. All countries in the Region reported a decline [7]. The variations in the incidence in different countries are depicted in Table 1.

While the reported incidence has increased, deaths due to measles are estimated to have decreased by 96% (from 255,133 to 9542) from 2013 to 2022. The significant enhancement in surveillance performance resulting in better reporting could be partly attributed to this observed phenomenon, as well as the shift in the age range of measles cases, which may have resulted in decreased number of deaths due to measles in the older age groups. In 2013, the Region contributed 26% of estimated global measles deaths [20]. This percentage declined to 7% (9542 of total 136,216) in 2022 [21,22,23].

### 3.3. Immunization Coverage

In 2013, all eleven countries in the Region had introduced a dose of measles-containing vaccine, while nine countries had introduced two doses in their routine childhood immunization program. Five countries had introduced one dose of rubella-containing vaccine, and three countries had introduced two doses of rubella-containing vaccines in 2013. By 2023, all countries in the Region had introduced two doses of a measles and rubella-containing vaccine in routine immunization [20] (Table 2).

The WHO-UNICEF estimated regional coverage for MCV1 increased from 85% in 2013 to an all-time high of 94% in 2019 [19] Similarly, the estimated regional coverage for MCV2 was 83% in 2019 compared to 59% in 2013, while the regional coverage of RCV1 was estimated at 93% in 2019 compared to 12% in 2013 [19] In 2020–2021, after the COVID-19 pandemic, the region observed a significant decline in coverages of measles and rubella-containing vaccines (Figure 1 and Figure 2). An estimated decline of 8% in MCV1 coverage, 11% in MCV2 coverage, and 7% in RCV1 coverage was witnessed in 2021 compared with 2019 levels. However, intensive efforts to catch-up, restore, and strengthen immunization in all countries ensured that coverage with MCV and RCV improved in 2022, achieving an estimated 92% coverage for MCV1, 85% for MCV2, and 92% for RCV1. Catch-up, restore, and strengthen efforts to improve coverage through routine immunization conducted by all countries resulted in an increase in routine coverage and have continued through 2023 and 2024.

There remain some variations in the timing of administration of MCV2 and RCV1 as well as of the coverage of MCV1, MCV2, and RCV in countries of the Region, as depicted in Table 3 below.

Based on reports received by national programs through the joint reporting mechanism [7], 322 million children were reported to have received MCV1, around 273 million MCV2, and 196 million children were reported to have received RCV1 as part of routine immunization activities between 2013 and 2023 [7]. More than 789 million children of various age groups received at least one additional dose of measles and rubella-containing vaccine through supplementary immunization activities (SIAs) during 2013–2022 [25].

### 3.4. Surveillance Performance

In 2013, following a regional resolution to eliminate measles and control rubella [26], all countries adopted laboratory-supported acute fever and rash surveillance and gradually moved away from outbreak-based surveillance; this transition was complete by 2022. This transition was also supported by an expansion of the laboratory network, increasing the number of WHO-proficient laboratories for measles and rubella surveillance from 23 in 2013 to 58 in 2023. Of these, 26 laboratories conduct both serology and viral detection for measles and rubella, and four laboratories also conduct viral genome sequencing. The sensitivity of fever and rash surveillance for measles and rubella, as measured by the key indicator of non-measles non-rubella discarded cases (A suspected case of measles/rubella that has been investigated and discarded as not being measles or rubella is termed a non-measles, nonrubella case if any of the following is true: An adequate specimen collected during the proper time period after the onset of rash has tested negative in a proficient laboratory; there is an epidemiological linkage to another communicable disease outbreak that is not measles or rubella; another etiology has been confirmed, regardless of whether it meets the definition of an epidemiological linkage; and the case does not match the definition of a clinically compatible measles/rubella case.) rate per 100,000 population increased from less than 0.5 per 100,000 population in 2013 to 5.13 per 100,000 population in 2023 [27]—a more than ten-fold increase in the sensitivity of surveillance [28]. Similarly, other surveillance performance indicators were maintained at high levels in 2023, although variations exist within countries and between countries (Table 4). A total of 207,155 suspected measles cases were reported in the Region in 2023, compared to 31,091 in 2020 [7].

### 3.5. Genotypes of Measles and Rubella Viruses

During 2013–2023, the laboratories in the Region reported B3, D4, D8, D9, and H1 genotypes of the measles virus, which are now on a decline. The transmission of D9 and H1 genotypes has not been reported since 2019 and of D4 since 2020. The only genotypes of measles reported from the Region in 2022 and 2023 are B3 and D8. For rubella, the genotype data are sparse, and there has been no reporting of any genotypes of rubella since 2021.The last reported rubella genotype was 1E from Thailand in 2020 (presented to the WHO South-East Asia Regional Immunization Technical Advisory Group meeting in August 2023).

To assist countries in establishing and implementing elimination standard surveillance for measles and rubella, a surveillance guide for measles, rubella, and congenital rubella syndrome was developed in 2017 [29] and revised in 2022 [30].

### 3.6. Epidemiological Analysis of Measles and Rubella Cases

The vaccination history of suspected cases of measles from surveillance data for various age groups shows the challenge in collecting accurate information on vaccination history, with many cases being reported as having unknown vaccination history.

The reported cases over the last five years indicate a shift in age group towards older age for measles but not for rubella (Figure 3).

Surveillance data of the last five years continue to show a significant number of reported cases of measles with unknown vaccination status, not only in the adult population but also in younger age groups, indicating challenges with recording and reporting. However, the number of zero-dose children over the last five years has decreased in younger age groups, indicating the effect of better routine vaccination as well as SIAs and catch-up vaccination efforts (Figure 4).

## 4. Discussion

The WHO South-East Asia Region has made significant strides toward elimination of measles and rubella [13]. Measles and rubella elimination was prioritized as a regional flagship program to support the countries to pursue the goal [31,32].

The region is home to almost a quarter of world population and a review of Demographic and Health Surveys from countries in the Region was conducted by WHO has shown that measles and rubella vaccinations are positively influenced by the education of mother (17.9 percentage point difference between children born to a mother with no education and higher education), education of father (16.6 percentage point difference between children born to a father with no education and higher education), place of delivery (14.7 percentage point difference between children born in health facility and elsewhere), birth order (14.1 percentage point difference between first-born children and 4+ birth order), household wealth quintile (14.7 percentage point difference between lowest and highest quintile) and number of antenatal care (ANC) visits (12.1 difference for measles between mother having 4+ visits and less than four visits). The region has been able to make progress in these social determinants, as demonstrated by the WHO study on review of the trends of demographic and health surveys over years [33].

This improvement in sociodemographic determinants may have enhanced access to vaccination with two doses of measles and rubella vaccine over the past decade, in addition to the introduction of MCV2, periodic SIAs, and catch-up vaccination conducted in the Region. These factors will need further enhancement to achieve very high coverage of two doses of measles and rubella-containing vaccine.

The surveillance performance in the Region as a whole has significantly improved, with all countries reporting immunization and surveillance data consistently at high rates despite diverse economic and social status.

While the reported number of cases has increased over the decade, the estimated deaths have decreased. This could possibly be due to better reporting due to enhanced surveillance performance as well as better case management, which could not be assessed in the article.

The reported genotypic information from the past several years indicates a decrease in the transmission of the number of genotypes of measles and rubella.

The implementation of strategies to eliminate measles and rubella had a jumpstart. The Region’s successes against polio and maternal and neonatal tetanus laid the foundations for building confidence to establish the goal of elimination of measles and rubella [34,35]. The infrastructure and capacity built previously for other vaccine preventable disease elimination, notably polio elimination, were re-purposed to focus on measles and rubella elimination [36,37,38].

The political commitment of countries of the Region to protect their populations from these deadly preventable diseases was evident by the regional resolutions [2], especially in the context where there is no global-level goal to guide the program and no concrete global partner support to pursue the goal further [39,40]. Identifying and declaring measles and rubella elimination as a flagship program by the South-East Asia Region underscored the commitment and gave additional impetus to accelerate progress [30,31]. The Region successfully rode on polio elimination initiatives to eliminate measles, as well as used the platform of measles elimination to eliminate rubella due to its programmatic feasibility [34,41].

The strategic plan for measles elimination and rubella and congenital rubella syndrome control in the South-East Asia Region for the period 2014-2020 [42] had a focus to develop the platform and systems required for measles and rubella elimination, while the strategic plan for measles and rubella elimination for the period 2020–2024 [4] provided the strategic framework to the program focused more on innovative approaches to accelerating progress.

The progress was built on a consultative process backed by evidence. Prior to each goal being set or rescheduled, an independent assessment was conducted, and countries and experts were consulted to ensure technical, programmatic, and political support [43,44]. Periodic evaluations at both the regional and national levels were prioritized in order to facilitate prompt course corrections [39,45,46]. This included the regular review of progress by the SEA-RVC and National Verification Committees (NVCs) for the elimination of measles and rubella [10,11,12,13].

The advocacy and the evaluations resulted in periodic vaccine introductions, along with improved routine services, supplemental immunization activities, and periodic intensification of routine immunization activities, considerably expanding access to MCV1, MCV2, and RCV1 between 2013 and 2022.

The SEA Region has made some progress towards measles and rubella elimination, some of it is attributed to the power of collaborations that brought together a remarkable spectrum of stakeholders at the national and subnational levels, including those beyond the health sector, to implement the strategies targeted towards achieving the goal [32,33,34,47]. The subnational accountability mechanisms created for the polio program in larger countries were critical collaborators to ensure implementation of strategies for measles and rubella elimination and were re-energized and given the responsibility to enhance implementation of the strategies to eliminate measles and rubella [48,49].

The contributions of the Regional and National Immunization Technical Advisory Groups to support acceleration of progress through overall technical leadership to the elimination program have remained noteworthy [15,16]. Similarly, the national verification committee (NVC) in each country collaborated with the relevant governments to oversee advancements made toward elimination of measles and rubella and provided annual reports to SEA-RVC and guided the program forward [10,11,12,13].

### Challenges

All WHO Regions have a regional mandate from Member States to pursue measles elimination, while the global program remains muted and does not have a global goal for measles and rubella eradication. As a result, the program is not able to harness the power of donors and partners to generate adequate resources to support optimal implementation of the strategies to eliminate measles and rubella.

The COVID-19 pandemic posed serious obstacles to achieving the target by 2023 and had a detrimental effect on vaccination and surveillance efforts. Despite the challenges brought on by the pandemic, Bhutan, DPR Korea, the Maldives, Sri Lanka, and Timor-Leste achieved and managed to sustain their measles and rubella elimination status. These countries will need continued efforts to sustain the elimination status as long as the measles and rubella viruses are circulating in neighboring countries and globally, by developing and effectively implementing the post-elimination sustainability plans. Larger countries like India and Indonesia have shown an increase in the number of measles cases, which may be partly because of the immunity gap added to the chronic gap during the pandemic and partly due to enhanced surveillance. These immunity gaps will need to be urgently addressed through tailored subnational targeted supplementary immunization activities to ensure these countries are back on track to achieve the measles and rubella elimination target.

There are several other challenges, the most significant of which is to achieve more than 95% coverage with two doses of measles and rubella-containing vaccine [50]. The subnational heterogeneity in coverage in several countries remains a substantial challenge and has resulted in outbreaks of measles in pockets of low coverage during recent years. The sensitivity of the surveillance for measles and rubella continued to remain sub-par in a large number of subnational areas in endemic countries, resulting in under-reporting and underestimating the disease burden as well as delaying a response to outbreaks. Adequate resources to optimally implement the strategies, coupled with national commitment not trickling down to the local level for optimal implementation of key strategies, have been identified as some of the other challenges to achieving measles and rubella elimination [51].

The delay in closing the immunity gap in children through high vaccination coverage has resulted in children escaping childhood without immunity to measles, which has led to a shift in the age group of the measles cases, making the elimination process more complex as this will require closing immunity gaps in other non-conventional age groups, increasing the cost of elimination.

## 5. Conclusions

While significant progress has been made across the Region on measles and rubella elimination, the region continues to be off track to achieve the elimination target by 2023 [44].

The year 2026 was considered to be the next feasible target date for the elimination of measles and rubella following a regional consultation with countries and experts that was held in 2023 [51]. The consultation considered the current momentum for measles and rubella elimination in the Region, the strengths, and enablers currently available across countries in the Region, the results from a modeling exercise, and factored in the loss of 2–3 years in program performance due to the COVID-19 pandemic for the achievement of the current target of measles and rubella elimination [50].

Measles and rubella elimination is feasible; however, this will require an optimal implementation of strategies within a very short period of time to achieve the target, which in turn is possible through significant public support and political will as well as sufficient financial resources mobilized for the program.

The year 2026 is an ambitious target to achieve measles and rubella in the WHO South-East Asia Region, which has a quarter of the world’s population. The WHO South-East Asia Region has gained the required momentum and has established the necessary framework to eliminate measles and rubella. It will remain critical for the Region to ensure broad public support, political will, and adequate resources and mechanisms to rapidly accelerate implementation of strategies over the next few years to achieve the goal of eliminating both measles and rubella.

## Figures and Tables

**Figure 1 vaccines-12-01094-f001:**
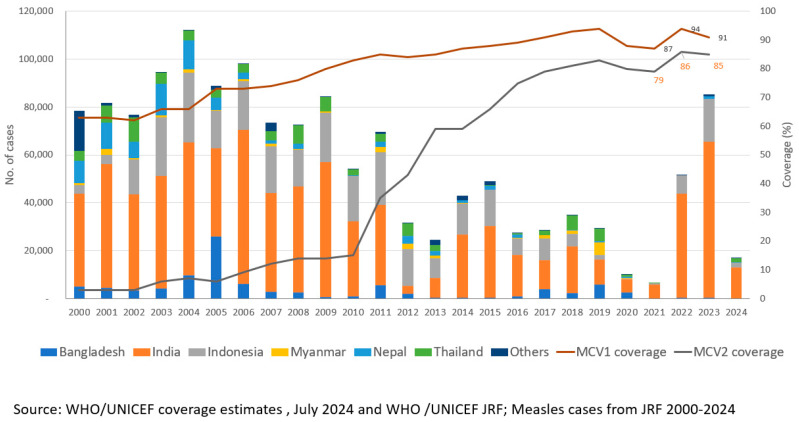
Number of reported measles cases * and estimated percentage of children who received their first and second dose of measles-containing vaccine (MCV), † by country—World Health Organization (WHO) South-East Asia Region (SEAR), 2000–2023. Abbreviations: MCV1 = first dose of MCV in routine immunization; MCV2 = second dose of MCV in routine immunization. * Cases of measles reported to WHO and the United Nations Children’s Fund (UNICEF) through the Joint Reporting Form from WHO-SEAR. † Data are from WHO and UNICEF estimates for SEAR, available at http://immunizationdata.who.int. Adapted from [24].

**Figure 2 vaccines-12-01094-f002:**
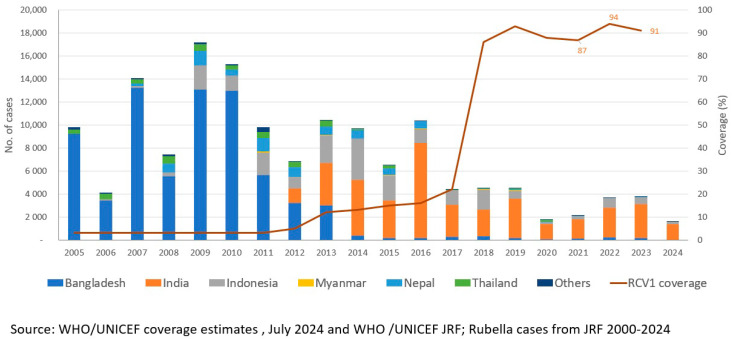
Number of reported rubella cases * and estimated percentage of children who received their first dose of rubella vaccine (RCV), † by country—World Health Organization (WHO) South-East Asia Region (SEAR), 2005–2023. Abbreviations: MCV1 = first dose of MCV in routine immunization; MCV2 = second dose of MCV in routine immunization. * Cases of measles reported to WHO and the United Nations Children’s Fund (UNICEF) through the Joint Reporting Form from WHO-SEAR. † Data are from WHO and UNICEF estimates for SEAR, available at http://immunizationdata.who.int. Adapted from [24].

**Figure 3 vaccines-12-01094-f003:**
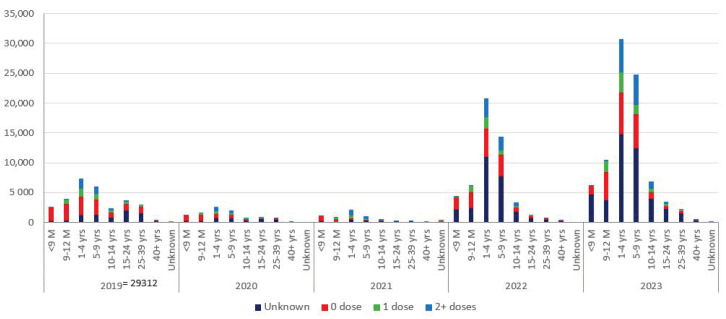
Reported cases of measles with vaccination status, WHO South-East Asia Region, 2019 to 2023.

**Figure 4 vaccines-12-01094-f004:**
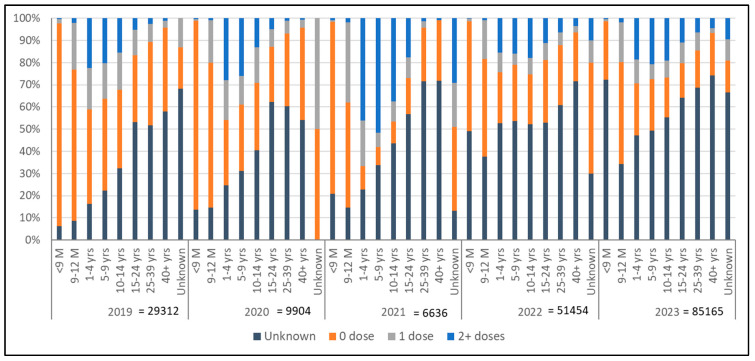
Proportion of measles cases, their vaccination status and age distribution, WHO Regional Office for South-East Asia, 2019 to 2023.

**Table 1 vaccines-12-01094-t001:** Reported cases and incidence of measles and rubella in WHO South-East Asia Region, by country, 2013 and 2023.

	2013				2023				% Change in Incidence	
	No. Reported Measles Cases (JRF †)	Measles Incidence §	No. Reported Rubella Cases (JRF †)	Rubella Incidence §	No. Reported Measles Cases (JRF †)	Measles Incidence §	No. Reported Rubella Cases (JRF †)	Rubella Incidence §	Measles	Rubella
Bangladesh	237	1.5	3034	19.7	281	1.6	182	1.1	6.7%	−94.0%
Bhutan	0	0.0	6	8.2	10	12.7	4	5.1	0.0%	−33.3%
DPR Korea	0	0.0	0	0.0	0	0.0	0	0.0	0.0%	0.0%
India	13,822	11.1	3698	2.9	65,150	45.6	2952	2.1	310.8%	−20.2%
Indonesia	8419	33.9	2355	9.3	18,063	65.1	584	2.1	92.0%	−75.2%
Maldives	0	0.0	0	0.0	5	9.6	0	0.0	0.0%	0.0%
Myanmar	1010	16.2	23	0.5	15	0.3	2	0.0	−98.1%	−91.3%
Nepal	1861	68.3	755	27.6	963	31.2	34	1.1	−54.3%	−95.5%
Sri Lanka	2107	102.9	24	1.1	810	37.0	0	0.0	−64.0%	−100.0%
Thailand	2641	40.7	539	7.7	64	0.9	12	0.2	−97.8%	−97.8%
Timor-Leste	4	3.4	0	0.0	7	5.1	3	2.2	50.0%	0.0%
Region Overall	30,101	16.2	10,434	5.5	49,201	40.9	3773	1.8	152.5%	−63.8%

Abbreviations: JRF = Joint Reporting Form; Data source: Data are from WHO and UNICEF estimates, 2023 revision (as of June 2024). Data available at http://immunizationdata.who.int (accessed on 18 July 2024); † JRF was submitted to WHO and UNICEF by member states with the official immunization data and the number of measles cases in the country for the year. § Incidence is calculated based on the reported cases and population by member states through WHO/UNICEF JRF.

**Table 2 vaccines-12-01094-t002:** Date of introduction of measles and rubella-containing vaccines in countries of WHO South-East Asia Region (SEAR).

Country	MCV1 †	MCV2 †	RCV1 †	RCV2 †
Bangladesh	1979	2012	2012	2015
Bhutan	1979	2006	2006	2006
DPR Korea	1997	2008	2019	2019
India	1985	2010	2017	2017
Indonesia	1982	2003	2017	2017
Maldives	1983	2007	2007	2017
Myanmar	1987	2012	2015	2017
Nepal	1988	2015	2013 end	2015
Sri Lanka	1984	2001	1996	2001
Thailand	1984	1996	1986	1997
Timor-Leste	1982	2016	2016	2016

† Abbreviations: MCV1—first dose of measles-containing vaccine; MCV2—second dose of measles-containing vaccine; RCV1—first dose of rubella-containing vaccine and RCV2—second dose of rubella-containing vaccine. Data Source: Immunizaitondata.who.int.

**Table 3 vaccines-12-01094-t003:** Estimated coverage with the first and second dose of measles-containing vaccine (MCV) and one dose of rubella-containing vaccine, vaccination schedule, by country—World Health Organization South-East Asia Region, 2013 and 2022.

Country	2013					2023					% Change, 2013–2023		
	WHO/UNICEF Estimated Coverage * (%)			MCV Schedule †		WHO/UNICEF Estimated Coverage * (%)			MCV Schedule †		MCV1 Coverage	MCV2 Coverage	RCV1 Coverage
	MCV1	MCV2	RCV1	MCV1	MCV2	MCV1	MCV2	RCV1	MCV1	MCV2			
Bangladesh	91	82	91	MR-9 m	M-15 m	97	93	97	MR-9 m	MR-15 m	7	13	7
Bhutan	94	89	94	MR-9 m	M-24 m	99	97	99	MMR-9 m	MMR-24 m	5	9.0	5
DPR Korea	99	76	§	M-9 m	M-15 m	28	16	28	MR-9 m	MR-15 m	−72	−79	280
India	83	51	§	M-9 m	M-16-24 m	93	90	93	MR-9 m	MR-16–24 m	12	76	930.0
Indonesia	87	76	§	M-9 m	M-6 years	82	62	82	MR-9 m	MR-18 m	−6	−18	820.0
Maldives	99	99	99	M-9 m	MMR-18 m	99	99	99	MR-9 m	MMR-18 m	0	0	0
Myanmar	86	80	§	M-9 m	M-18 m	74	65	74	MR-9 m	MR-18 m	−14	−19	740
Nepal	88	0	88	MR-9 m	§	93	89	93	MR-9 m	MR-15 m	6	100.0	6
Sri Lanka	99	99	99	MMR- 12 m	MMR 3 years	99	99	99	MMR-1 y	MMR-3 y	0.0	0	0.0
Thailand	99	94	99	MMR-9 m	MMR-7 y	93	87	93	MMR-9 m	MMR-2.5 y	-6	-7	-6
Timor-Leste	70	0	§	M-9 m	§	72	72	72	MR-9 m	MR-18 m	3	100.0	720
SEA Region	78	53	12			91	85	91			17	60	658

Abbreviations: m = months; M = measles; MCV = measles-containing vaccine; MR = measles-rubella; MMR = measles-mumps-rubella; UNICEF = United Nations Children’s Fund; y = years. * Data are from WHO and UNICEF estimates, 2023 revision (as of 15 July 2024). Data available at http://immunizationdata.who.int. † As reported to WHO/UNICEF on JRFs for the year. § vaccine was not introduced into routine immunization.

**Table 4 vaccines-12-01094-t004:** Key surveillance performance indicators in SEAR countries, 2023.

2023	Number of Suspected Measles Cases	% Suspected Cases with Adequate Investigation-Initiated w/in 48 h of Notification	% Suspected Cases with Serum Specimens Collected	Number of Serum Specimens Received in Laboratory	% Serum Specimens Received at Laboratory w/in 5 Days of Collection	% Specimens Tested for Serology of Specimens Received in the Laboratory	% Serology Results Reported w/in 4 Days of Specimen Received at Laboratory	Non Measles, Non-Rubella Discard Rate per 100,000 Population	% Districts with Non-Measles Non Rubella Discard Rate of 2/100,000 Population
Bangladesh	7346	97.2	89.4	6529	99.5	100.0	97.5	3.92	100
Bhutan	229	94.0	94.8	181	85.4	100.0	65.4	28.43	80
DPR Korea	491	100.0	100.0	491	100.0	100.0	100.0	1.99	No data
India	154,475	73.4	77.0	116,105	100.0	100.0	82.9	5.75	94
Indonesia	39,140	77.8	75.8	21,508	97.6	100.0	12.9	4.84	55
Maldives	177	73.4	85.3	149	100.0	100.0	84.7	28.97	62
Myanmar	180	100.0	98.9	174	100.0	100.0	100.0	0.3	5
Nepal	3030	98.9	81.7	2469	100.0	100.0	76.6	6.94	90
Sri Lanka	1217	76.5	80.4	1159	99.3	100.0	62.3	1.7	19
Thailand	668	54.3	90.4	569	93.1	100.0	95.8	0.94	No data
Timor-Leste	202	100.0	97.8	198	100.0	100.0	37.3	13.99	62
SEAR	207,155	75.6	77.5	149,532	99.6	100.0	73.7	5.13	

## Data Availability

The data presented in the study are openly available at https://immunizationdata.who.int/ and https://www.who.int/southeastasia/health-topics/immunization/vaccine-preventable-disease-(vpd)-surveillance-data (accessed on 18 July 2024).

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
