# Peer review of "Progress towards Measles and Rubella Elimination in the South-East Asia Region—2013–2023"

_vaccines, 2024, doi:10.3390/vaccines12101094_

Round 1

Reviewer 1 Report

Comments and Suggestions for Authors

The authors provide an assessment of the measles and rubella eradication programme in the South-East Asia region. This is a valuable document, well written and with adequate tables and figures. The effort made is commendable, but it will be difficult to achieve the objectives by 2026. Eradication programmes have been complicated by the COVID-19 pandemic, but there are other factors that make control very difficult, especially in highly populated countries.

Author Response

Author response:

Thank you for the comment . Agree with the comments , rows 209 to 226 have been added to describe other determinants . Similarly , row 333 have been added to highlight the issue described above.

Thank you

Reviewer 2 Report

Comments and Suggestions for Authors

This study is a review paper on the activities and outcomes of the WHO South-East Asia Region's efforts to address measles and rubella. The data from various countries have been extensively and well-organized, and since this is a review paper, there is little need for revision.

  1. Since the situations of the countries included in the SEA region are very diverse, and their economic scales, development patterns, and reporting rates of data differ, it would be necessary to mention this point in the discussion section.
  2. In the case of DPR Korea, the data suggests that there are no cases of measles, but given the limitations of the available data, this needs to be verified.
  3. in methods, the authors mentions that they used data from 2022 when 2023 data were unavailable. It would be beneficial to clarify how this might impact the accuracy of the conclusions drawn.

4.Please correct any editorial errors in the content  (page 2, line 82, page 3 line 103 etc..)

Author Response

RESPONSE: Thank you for the comments , these will add value to the  article and have given additional insights tothe authors. 

response to point 1. Findings related to demographic and social determinat of measles vaccianiton have been added  starting line 208 to line 220

response to point 2. the data quoted forDPRKorea are submitted by WHO and UNICEF countrry offices in DPRKorea  through eJRF after joint review , author have cross checked with WHO and UNICEF country offices again . 

response to point 3. Most for 2023 are now publicly available after July 15 2024 and thus data have been updated for 2023 in Table 2 which had 2022 data earlier. Mortality estimates ( line 82) for 2023 are not available and  are not expected to have any impact on the conclusions drawn

response to point 4. Editorial corrections done.

Thankyou

Reviewer 3 Report

Comments and Suggestions for Authors

The paper is a review on measles and rubella incidence and vaccination in 11 South-East Asian countries over the 2013-2023 period. The study is hardly more than an update on references 1-5. The paper shows that the countries had achieved much reduction in measles and rubella incidence between 2013 and 2019 through mass vaccination, but because of Covid-19 vaccine coverage declined, and incidence increased again. The paper is part of a special issue on measles elimination.

The paper is essentially descriptive, well documented, clearly written. It will fit in the special issue, and will be useful for readers. It could be improved before publication (typos, tables, figures).

Comments:

1) The paper makes little use of geography and population size: 5 of the 11 countries that achieved measles elimination by 2023 are small and isolated countries (3 islands, 1 mountainous, 1 politically isolated). The 2 very large countries (India and Indonesia) had increasing incidence. A note on this point would be welcome.

2) A separate and more detailed analysis of India and Indonesia could be useful, as most cases in 2022-2023 come from these countries.

3) No critical analysis on the quality of data sources: how were measles deaths estimated? For example, DPR-Korea reports no case of measles or rubella neither in 2013 nor in 2023. How can this be? A note on data quality and reliability would be useful.

4) How to reconcile the fact that measles incidence increased while measles mortality declined? Is this an age effect? Or due to misreporting? Is this true in all six countries? More details on this point would be welcome.

5) A formal analysis of the distribution of cases by vaccination status would be most welcome. Is there a real change over time? Did the second dose added significant protection or change in the disease dynamics?

Details / Typos

Line 32: “resolved to eliminate both measles and rubella (defined as for measles) by 2023”

            It seems that (defined as for measles) is useless. Delete? A footnote defining “elimination” could be useful.

Line 105: compared to 12% in 2013.19

            What is 19? Is it a reference? [19]

Line 163: “.13 per 100 162 000 population in 202328 –“

What is 28? Is it a reference? [28]

Line 171: “WHO South-East Asia Region (SEAR) has eleven Member States – Bangladesh, Bhutan, DPR Korea, India, Indonesia, Maldives, Myanmar, Nepal, Sri Lanka, Thailand, Timor-Leste;

            This should be said from the beginning; no need to repeat.

Line 210-214: This could also be due to a change in the age at onset. (measles case fatality declines very quickly with age).

Line 215-218: Poorly stated. Was there a change in the circulating serotypes or genotypes? Does this have any implication for transmission?

Page 5-6: The footnote is split over two pages (hard to read).

Line 303: 52 is probably a reference [52].

Tables and Figures

Table 1: in Bhutan the number of measles cases was 0 in 2013 and 10 in 2023. How can this make no increase (0%) ? If increase cannot be computed because of 0 case at baseline, just say “NA” or “-”.

Table 2: Vaccine coverage in DPR Korea is written 0 in 2023, whereas these are most likely missing values. Just say “NA”.

            If vaccine was not available, just say 0% (instead of §)

            Calculations of changes are often wrong or misleading. For instance, if (97 – 91)= 6.6%, how 95 – 0 makes 950.0%? Best would be to use the same format with 1 decimal (99.9%), or 0 if not vaccination, or “NA” is not available.

Table 4: This table could be put at the beginning of the Results section, as this information is important to understand the rest [optional].

Figures 3 and 4: figure are poorly presented and hard to read. It would read better if each year was represented as a single bar (adding to 100%), with all the age groups, and a line linking corresponding age groups. So readers could see which age group increased.

Or, the figure could be replaced by a Table.

Or, the figure could be split into two figures: one showing the changing age distribution (single bar per year), and one showing the changing distribution by number of doses (in absolute value).

            Authors could also compute the mean age at onset, by vaccination status, and compare trends over time (would be most useful).

[Optional] An annex table with the number of cases and deaths by country and year, and proportion vaccinated would be useful.

Author Response

Author response:

Thank you for the valuable insights

Response to comment 1- rows 293 to 301 added in the paper to acknowledge the issue and the need for urgent targeted approach in large countries

Response to comment 2-partly responded  in rows 293 and 301, considering the importance,   a separate publication on India is in the process of being finalized and published,

Response to comment 3- these are important issues highlighted by the reviewers. The data taken are from secondary public available source and from joint reporting of WHO and UNICEF from each od these countries and thus the article did not dwell much on these.

Response to comment 4- row 84 to 86 added to address this . The surveillance data shows shift in age of measles to older age groups and significant enhancement in surveillance performance resulting   in a  better reporting could be partly attributed to this observed phenomenon.

Response to comment 5-  this is wonderful idea provided to the author. Attempt to address vaccination status was made in figure 3 , the author plans to have another detailed analysis of the effect of MCV2 introduction in SEAR and publish. Considering that the article is focused on progress made, this has been limited to general review of cases and vaccination status pattern.

Details / Typos

Line 32: “resolved to eliminate both measles and rubella (defined as for measles) by 2023”

            It seems that (defined as for measles) is useless. Delete? A footnote defining “elimination” could be useful.

The line within brackets hav been removed. Foot note on defining elimination available  in first page

Line 105: compared to 12% in 2013.19

            What is 19? Is it a reference? [19]

Addressed, it’s a reference

Line 163: “.13 per 100 162 000 population in 202328 –“

What is 28? Is it a reference? [28]

Addressed, it’s a reference

Line 171: “WHO South-East Asia Region (SEAR) has eleven Member States – Bangladesh, Bhutan, DPR Korea, India, Indonesia, Maldives, Myanmar, Nepal, Sri Lanka, Thailand, Timor-Leste;

            This should be said from the beginning; no need to repeat.

Deleted from here and added to footnote in first page

Line 210-214: This could also be due to a change in the age at onset. (measles case fatality declines very quickly with age).

Fully agree, lines 84-85 added to address this

Line 215-218: Poorly stated. Was there a change in the circulating serotypes or genotypes? Does this have any implication for transmission? (now lines 235)

This implies that the transmission has reduced and several genotypes are no longer seen  and to emphasis the need to have genotype  information  as one of measure of progress towards elimination. Second part of the line deleted for clarity.

Page 5-6: The footnote is split over two pages (hard to read).

Formatting issue  addressed

Line 303: 52 is probably a reference [52].

Addressed, it’s a reference

Tables and Figures

Table 1: in Bhutan the number of measles cases was 0 in 2013 and 10 in 2023. How can this make no increase (0%) ? If increase cannot be computed because of 0 case at baseline, just say “NA” or “-”.

Revised

Table 2: Vaccine coverage in DPR Korea is written 0 in 2023, whereas these are most likely missing values. Just say “NA”.

            If vaccine was not available, just say 0% (instead of §)

The country had not introduced RCV  in 2013 thus § was kept to indicate this in foot note.

            Calculations of changes are often wrong or misleading. For instance, if (97 – 91)= 6.6%, how 95 – 0 makes 950.0%? Best would be to use the same format with 1 decimal (99.9%), or 0 if not vaccination, or “NA” is not available.

Addressed. These were used as percent decrease and not in absolute point percent .

Table 4: This table could be put at the beginning of the Results section, as this information is important to understand the rest [optional].

Thank you

Figures 3 and 4: figure are poorly presented and hard to read. It would read better if each year was represented as a single bar (adding to 100%), with all the age groups, and a line linking corresponding age groups. So readers could see which age group increased.

Or, the figure could be replaced by a Table.

Or, the figure could be split into two figures: one showing the changing age distribution (single bar per year), and one showing the changing distribution by number of doses (in absolute value).

            Authors could also compute the mean age at onset, by vaccination status, and compare trends over time (would be most useful).

[Optional] An annex table with the number of cases and deaths by country and year, and proportion vaccinated would be useful.

Revised figures submitted for more clarity